# Long-term genomic coevolution of host-parasite interaction in the natural environment

Elina Laanto [1], Ville Hoikkala[1], Janne Ravantti[2] & Lotta-Riina Sundberg [1]

Antagonistic coevolution of parasite infectivity and host resistance may alter the biological functionality of species, yet these dynamics in nature are still poorly understood. Here we show the molecular details of a long-term phage–bacterium arms race in the environment. Bacteria (*Flavobacterium columnare*) are generally resistant to phages from the past and susceptible to phages isolated in years after bacterial isolation. Bacterial resistance selects for increased phage infectivity and host range, which is also associated with expansion of phage genome size. We identified two CRISPR loci in the bacterial host: a type II-C locus and a type VI-B locus. While maintaining a core set of conserved spacers, phage-matching spacers appear in the variable ends of both loci over time. The spacers mostly target the terminal end of the phage genomes, which also exhibit the most variation across time, resulting in arms-race-like changes in the protospacers of the coevolving phage population.

[1] Centre of Excellence in Biological Interactions, Department of Biological and Environmental Science and Nanoscience Center, University of Jyvaskyla, PO box 35, FI-40014 Jyvaskyla, Finland. [2] Department of Biosciences and Institute of Biotechnology, Viikinkaari 9, University of Helsinki, FI-00014 Helsinki, Finland. Correspondence and requests for materials should be addressed to L.-R.S. (email: lotta-riina.sundberg@jyu.fi)

One of the fundamental questions in evolutionary biology and medicine is how host immunity drives the evolution of pathogens, as all living organisms are exposed to infections that can cause significant harm and selection in the host population. The antagonistic coevolutionary arms race of parasite infectivity and host resistance leads to adaptations and counter-adaptations in the coevolving partners[1–4], and also has a central role in the evolution of host–parasite relationships in the microbial world. This arms race has been shown especially under experimental settings, where lethal infections by bacterial viruses, (bacterio)phages, shape the diversity and dynamics of the coevolving host bacterial populations[1, 2, 4, 5], whereas the phages have the capacity to rapidly overcome host immunity[1, 5–9]. It has been suggested, however, that the adaptive potential of phage and high costs in phage resistance in bacteria can limit phage–bacterium coevolution[10]. Yet, the arms-race dynamics observed in laboratory experiments substantially differ from the real-life dynamics under the complex web of surrounding interactions present in the environment, which influences the ecology and evolution of both phages and their hosts[11, 12].

CRISPR (clustered regularly interspaced short palindromic repeats) and associated *cas* genes form the bacterial CRISPR-Cas adaptive immune system against phages and plasmids. CRISPR-Cas protects bacteria from infections by cleaving invading nucleic acid sequences (protospacers) that are identical or nearly identical to the spacers in the bacterial CRISPR repeat-spacer array[13, 14]. Protospacer adjacent motifs (PAMs) are used to distinguish self from non-self and are crucial in most CRISPR systems[15], while protospacer flanking sites (PFSs) determine the efficiency of interference in RNA-targeting CRISPR systems but have no known autoimmunity-related functions[16]. Over time, novel spacers accumulate in one end of the array and may therefore confer resistance to multiple phages. However, in accordance with antagonistic coevolution theory, phages can counter-adapt to host immunity via mutations in the protospacer or PAM/PFS sequences[5, 17, 18], or by using anti-CRISPR proteins[19–22].

Following evolutionary change in natural communities is challenging, and linking genomic change data with the coevolutionary dynamics of phages and bacteria is one of the key challenges in microbiology. Therefore, a comprehensive view of host–parasite coevolutionary dynamics in the environment, surrounded by a network of other trophic interactions, is missing. In fact, previous studies have concentrated either on phenotypic changes[7, 23] or CRISPR genetics using metagenomic approaches[24–26]. Particularly, empirical evidence on the relative importance of constitutive and adaptive resistance (mutations preventing successful phage life cycle and CRISPR loci insertions, respectively[27]) in bacteria and the corresponding changes in the parasitic phage[28] supported by phenotypic data from natural settings is missing. Constitutive and adaptive resistance mechanisms are predicted to be important in different ecological conditions[29–31], therefore teasing apart the relative role of different bacterial resistance mechanisms is essential for understanding microbial community ecology. From an applied perspective, such information is also crucial for the development phage therapy applications[32], where evolution of resistance is expected to limit the functionality of the treatment over time.

The importance and volume of aquaculture is growing steadily to meet the increasing need for high-quality protein for human consumption[33], and new methods such as phage therapy, to control and manage bacterial diseases are under vigorous research[34, 35]. Here, we characterize the coevolution of populations of bacteriophages and their bacterial hosts (i.e., the fish pathogen *Flavobacterium columnare*[36]) at the phenotypic and genetic level, in a flow-through aquaculture setting during the period 2007–2014. We sampled for bacterial and phage isolates

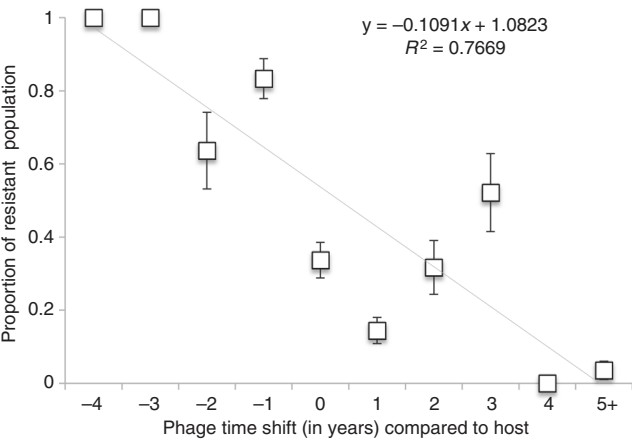

**Fig. 1** Time shift of phage–bacterium coevolution in the fish farming environment. Phage–bacterium coevolution measured as mean proportion (± S.E.) of resistant host population, described as annual relative change in host resistance when exposed to phages from the past (−4 to −1 years), contemporary (0), and future (+1 to +5 years). The bacteria are in general resistant against phages from the past but susceptible to infection by phages from contemporary and future time points (24% and 18% resistant, respectively; GLMM, $F_{(2, 507)} = 15.099$, $p < 0.001$)

from the fish farming facilities or immediate surroundings and performed all-against-all cross-infections to analyze bacterial resistance patterns. To link the phenotypic patterns with molecular changes, we sequenced phage genomes to understand the determinants of the host range and characterized the bacterial CRISPR loci to understand the role of adaptive immunity in driving genome evolution and host range in the phage population. We report the patterns of arms race coevolution in a natural community and demonstrate that whereas bacteria evolve resistance, phages evolve a broader host range over time, which is associated with increase in genome size. We also observed evolutionary change in phage genomes in response to bacterial adaptive (CRISPR) and constitutive immunity, exemplifying the importance of both resistance mechanisms in natural bacterial populations. We also demonstrate a type VI-B CRISPR system functioning in its natural host and in a natural setting.

## Results

**Phage and bacterial isolates**. *F. columnare* phages are genotype-specific, each infecting strains of only one genotype[37]. This specificity allowed us to analyze both phage and host isolates over long time scales in locally adapting populations. The phage and bacteria were collected from a fish farm in Central Finland mainly during the warm water season when outbreaks occur. During the period 2007–2013, we isolated 17 *F. columnare* isolates that belong to the genetic group C (Supplementary Table 1). Furthermore, we isolated 30 dsDNA phages (belonging to the family Myoviridae) during 2009–2014, which infect specifically this bacterial host group (Supplementary Table 2).

**Coevolution of phage infectivity and bacterial resistance**. First, we analyzed the phage host range by cross-infecting all 30 phage isolates with all 17 bacterial isolates. The most recently isolated phages had the widest host range, being able to infect nearly all bacterial hosts. Analyzed using a time-shift approach[38], the phage isolation time point had a significant effect on infectivity compared to the bacterial host. The bacteria were in general resistant against phages from the past but susceptible to infection by phages from contemporary and future time points (24% and 18%

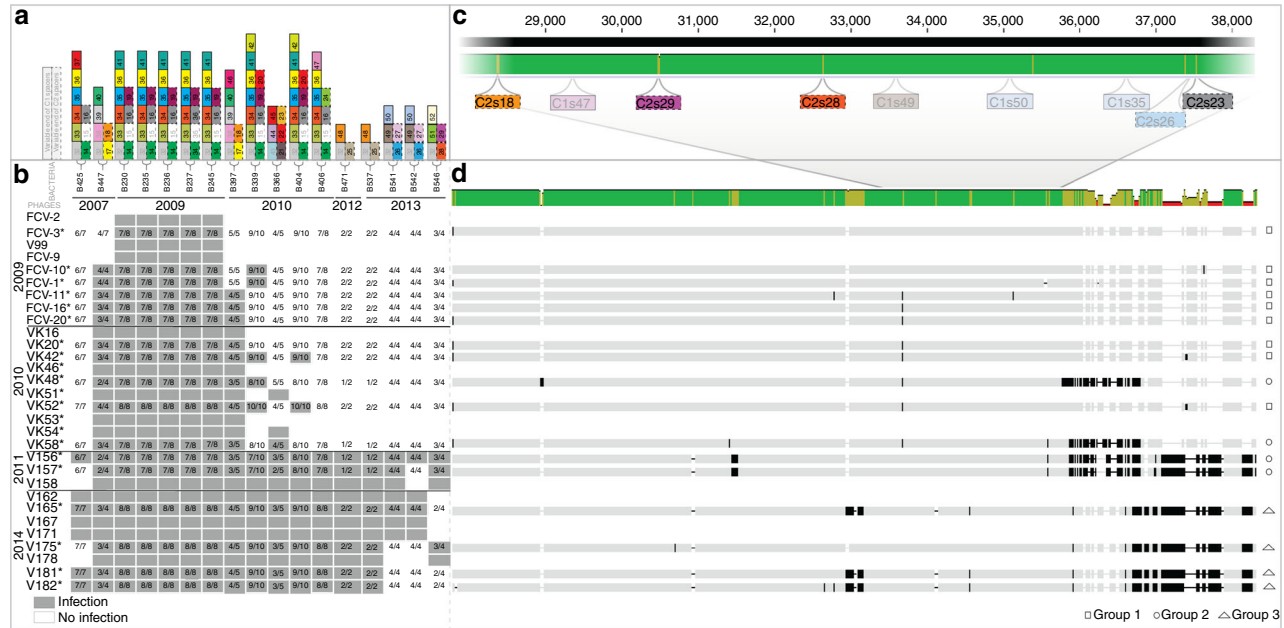

**Fig. 2** Arms race coevolution of phage infectivity and host resistance at genetic level. The effect of long-term genomic coevolution on phage infectivity and genome, and on host bacterium CRISPR content. **a** Spacer diversity in the variable ends of both CRISPR loci (C1 and C2) is indicated by numbered and colored *rectangles* above each bacterial strain (each number and color referring to a specific spacer within a locus). C1 spacers are marked with *solid lines* (left spacer column) and C2 spacers with *dotted lines* (right spacer column). Numbers and colors are locus-specific and only variable spacers are shown. All spacers except C1s32, C1s38, and C2s15 are targeting the phages. **b** Host range of phages (rows) infecting *Flavobacterium columnare* isolates (columns). *Gray rectangles* indicate infection (presence of plaques) and *white rectangles* indicate resistance. The first number within these *rectangles* indicates the number of CRISPR spacers with an identical match in the phage genome and the second number the maximum number of phage-targeting spacers the bacterium has. **c** Close-up: CRISPR protospacer regions mapped to a ~11 kb portion of the multiple sequence alignment, where the four highlighted protospacers indicate specific, possibly CRISPR-driven, changes in these areas. **d** Patterns of molecular evolution in the phage genomes in response to host range. Each sequenced phage genome follows the corresponding host range row from **b**. *Green color* in the consensus sequence (above) indicates identical sequence, *yellow* 30%, and *red* < 30% identity. Note that also the bar height changes accordingly. *Black* indicates nucleotide differences and insertion elements

resistant, respectively; generalized linear mixed model (GLMM), $F_{(2, 507)} = 15.099$, $p < 0.001$). The phage isolation year included as a random factor in this analysis had a significant effect (Wald $Z = 2.959$, $p = 0.003$), while bacterial isolation place within the farm (inlet water, outlet water, fish) or phage enrichment host did not ($Z = 1.36$, $p = 0.174$, $Z = 0.586$, $p = 0.558$, respectively). All bacteria were resistant to phages from 3–4 years from the past, and, on the other hand, bacterial resistance was below 0.5% against phages from 4–5 years in the future (Fig. 1). In addition, enrichment host did not affect the phage host range (number of infected hosts, Kruskal–Wallis $\chi^2 = 11.7$, df = 7, $p = 0.103$).

We tested phage adsorption efficiency of eight phages on nine bacteria isolated in different years (Supplementary Fig. 1) to see whether the role of cell surface modifications over other resistance mechanisms could be detected. Adsorption efficiency of the phages differed from 0 to 91%. In some cases, adsorption was less efficient in all or some of the resistant strains indicating the possibility of resistance via surface modifications and hence the lower adsorption efficiency for the phage. Also, increased adsorption was detected in some of the resistant strains (e.g., VK42 to B471 and VK58 to B406), which suggest post-adsorption resistance mechanisms in these pairs. In addition, the efficiency of plating (EOP), adjusted to the titers on strain B230 that was infected by all phages (Supplementary Table 3), did not relate to the observed adsorption.

When analyzed using generalized linear mixed models with adsorption percentage as the response variable, resistance was found to have no significant effect in adsorption. Phages

originating from a previous time point (as compared to the bacterium) were predicted to have reduced adsorption percentage (approximately one-third of that of contemporary phages, GLMM, $F_{(2, 212)} = 6.506$, $Z = -2.722$, $p = 0.006$). No significant difference in adsorption was observed between contemporary and future phages.

**Host resistance drives phage genome evolution**. To understand phenotypic resistance at the genomic level, we sequenced genomes of 17 phage isolates from 2009 to 2014. This resulted in complete genomes with lengths ranging from 46,481 to 49,084 bp (Fig. 2; Supplementary Table 2; Supplementary Fig. 2). Phage genomes from different years were highly similar, enabling us to study differences at the nucleotide level. From the predicted 74 (2009) to 76 (2014) open reading frames (ORFs), 52 were 100% identical between all genomes, including putative terminase, portal protein, and several structural proteins (e.g., major capsid protein). However, we observed non-synonymous differences in putative tail and structural proteins (two phages from 2011 in ORFs 27, 28, and three phages from 2014 in ORFs 36 and 37, Supplementary Fig. 2). This could indicate a conventional arms race between phage infectivity and host resistance via receptor mutation in natural settings (see refs [8, 9] for similar experimental evolution results).

In multiple sequence alignment, the phage genomes clustered in three major groups (Fig. 2, marked with symbols). Genomes of phages in Group 1 (seven phages from 2007 to 2009) were

identical. In Group 2 phages (four phages from 2010 to 2011), genomic changes were located at 38,922–43,395 bp (consensus sequence, in phage VK48 starting from 38,421 bp). According to Blastp analysis, this genomic region might contain ORFs associated with phage replication. Furthermore, these differences were missing in the Group 1 and Group 3 phages (Supplementary Discussion).

The broadest host range phages (Group 3 with four phages; isolated in 2014) had changes in putative structural proteins (ORFS 36 and 37, see above). Interestingly, they, as well as two phages isolated in 2011 from Group 2 had extra DNA at the terminal end of their genomes, coding additional predicted ORFs. All these additional ORFs are located upstream of ORF66, which has a putative ICE (integrative and conjugative element) domain. Together with changes in the bacterial CRISPR spacer content (see below), these genomic differences seem to correlate with a broader phage host range. The additional ORFs were assigned as hypothetical proteins. One of these ORFs (no. 67.1) has a conserved domain belonging to the family of N-acyltransferases (Supplementary Fig. 2; Supplementary Table 4). Another ORF (no 74.1) is a putative YopX protein, having also mobile and extrachromosomal element and prophage functions.

In addition, we found indications that phage infectivity may significantly differ in the three genomic groups (Groups 1, 2, and 3, indicated in Fig. 2). Infectivity was interpreted from EOP values (Supplementary Table 3; Kruskal–Wallis, $\chi^2 = 38.571$, df = 2, $p < 0.001$, Supplementary Fig. 3). However, it should be noted that here we were not able to test the plaque production against contemporary bacterial isolates, which may influence the results.

**Characterization of the CRISPR loci and PAM sequences**. Next, using a previously published *F. columnare* genome sequence[39] and our own sequence data, we identified two CRISPR loci found in all 17 *F. columnare* isolates used in this study, designated here as C1 (CRISPR1, ATCC49512: 391567–394479) and C2 (CRISPR2, ATCC49512: 1680008–1680571) (Supplementary Fig. 4; Supplementary Table 5). C1 contains Cas9 followed by a repeat-spacer array and two smaller *cas* genes (Cas1 and Cas2) in opposing directions (Supplementary Fig. 4). On the basis of *cas* gene composition[14] and Cas9 homology[40], C1 is a type II-C CRISPR locus. C2 represents the recently discovered type VI-B CRISPR system with its large RNA-targeting single-component endonuclease Cas13b. We confirm that Csx27 and Csx28, small *cas* gene regulators associated with some type VI-B systems, are missing from *F. columnare* as noted by Smargon et al.[41]. Two putative transposases were also discovered immediately downstream of this locus.

Repeat-spacer arrays from both loci were sequenced and corresponding protospacers searched in the phage genomes. Across all bacterial isolates, a total of 52 unique C1 spacers were discovered, 18 of which were targeting phage genomes in this study on both coding and non-coding strands (8 and 10 spacers, respectively), and both intergenic and predicted ORF regions (5 and 13 spacers, respectively). A total of 29 unique spacers were discovered in C2. Interestingly, all 15 C2 spacers are targeting predicted ORFs on the phage coding strand, supporting the notion that type VI-B systems may be targeting viral transcripts[41].

Protospacer distribution in the phage genomes was biased toward the terminal end of the genomes (Supplementary Fig. 5) and differed significantly from an expected uniform distribution in both loci (one-tailed permutation test, C1 $p = 0.001$, C2 $p = 0.016$). Roughly half of C1 protospacers were located in the fourth quadrant of the phage genomes, with C2 protospacers following a similar yet less pronounced trend.

Analyzing regions surrounding C1 protospacers revealed a putative downstream PAM sequence of NNNNNTAAAA (Supplementary Table 6). A similar analysis of C2 protospacer PFSs supported the findings of a previous study[41] and also implied possible involvement of C1 acquisition machinery in C2 spacer acquisition (Supplementary Table 7; Supplementary Discussion).

**Specificity of the phage–bacterium coevolution**. Over time, all bacterial strains displayed variation in spacer content while maintaining a large set of conserved spacers shared among all isolates. The variable ends did not only vary in sequence, but also in the number of spacers, with the surprising finding that the most recent bacteria had generally fewer spacers than older isolates (Fig. 2a). Whereas none of the conserved spacers were targeting phages, the variable ends of the arrays consisted almost solely of phage-targeting spacers (Fig. 2a; Supplementary Data 1), exemplifying how exclusively CRISPR evolution is driven by the phages in close proximity in time and space. Individual bacteria often contained multiple spacers matching individual phages (Supplementary Fig. 5), demonstrating the specificity of the host–parasite interaction. Furthermore, the incorporation of novel DNA in the ends of the Group 2 and 3 phage genomes was followed by the appearance of two CRISPR spacers targeting these areas (C2s22, C1s52). Surprisingly, however, one spacer (C2s16) was already present in the bacteria (Fig. 2a) before the emergence of a novel protospacer-containing sequence in the phages.

In addition to establishing that new bacterial spacers mostly originate from phages, we also considered possible coevolutionary dynamics between the observed spacer and protospacer sequences. Specifically, we wanted to see if phage protospacer or PAM/PFS sequences vary after the appearance of phage-targeting spacers in the bacterial population. Overall, 6 of the 18 phage-targeting C1 spacers and 9 of the 15 phage-targeting C2 spacers had altered protospacer sequences in phages isolated after the appearance of the corresponding spacer in the bacterial population (Supplementary Fig. 5). Some protospacer alterations resulted from large re-organizations in the terminal ends of the genomes, while some were due to very specific single-nucleotide polymorphisms or deletions within the protospacer sequences. Most specific sequence variations took place on a roughly 11 kbp area (27,352–38,174 in the consensus sequence). This area is targeted by four C1 and five C2 spacers (Fig. 2c; Supplementary Table 8), and includes six mutational sites (1–12 bp in length). Four of these mutational sites are located within the C2 protospacers (spacers C2s18, C2s23, C2s28, and C2s29) and their appearance correlates chronologically with the introduction of the corresponding spacers to the bacterial population (Supplementary Fig. 5). Following the appearance of these alternative proto-spacers, these spacers were no longer detected in later bacterial isolates. Interestingly, the ancestral protospacer sequence (targeted by spacers C2s18 and C2s23) reappeared in later phage isolates, which could suggest that bacterial strains harboring these spacers no longer co-exist.

Currently, knowledge on class 2 affiliated anti-CRISPR proteins is very limited. On the basis on the findings of a recent study[22], we searched for homologs of the anti-CRISPR proteins AcrIIA1, AcrIIA2, AcrIIA3, and AcrIIA4 from all the phage genomes in our data set. No significant hits were recovered.

Despite the apparent coevolutionary dynamics displayed by the CRISPR arrays and phage genomes, linking CRISPR data with phenotypic infection patterns did not, in most cases, reveal clear correlation between bacterial resistance and spacer content (Fig. 2b).

## Discussion

Our results highlight the long-term arms-race coevolution of hosts and parasites in an aquaculture environment down to the molecular level. In accordance with previous findings[1, 7–9], the bacterial population was generally resistant against phages from past time points and susceptible to infections by (broad host range) phages from the future. Further molecular characterization pinpointed genetic changes in both host and phage populations that fit the assumption of arms-race coevolution. This allows estimations of the role of constitutive and adaptive immunity as drivers of parasite infectivity in environmental populations. These results are especially important for economically relevant conditions that might be suitable for phage therapy approaches.

The evolution of generalist parasites has been shown to arise in response to host resistance in experimental settings[1, 42]. Combined with phenotypic resistance patterns, our data suggest that the evolution of host immunity may have caused directional selection on phage infectivity and host range via genomic expansion (Fig. 2; Supplementary Fig. 3). In correlation with higher infectivity (Supplementary Fig. 1), the genomes of the most recently isolated phages had larger genome size, increasing from 46,448 bp (in 2009) to 49,121 bp (in 2014). Phage host range was independent of the enrichment host used in the original phage isolation. An earlier isolated enrichment host was always used (except for first five phages in 2009 (Supplementary Table 2), however, similar phages could be isolated even when using different enrichment hosts (e.g., phage pairs VK156 and VK157, and FCV-10 and FCV-11).

All additional DNA was concentrated in the terminal regions of the genomes. Although no clear functions for the three additional ORFs were predicted, structural similarity to a YopX-like protein was found. Although this protein is an outer membrane protein of *Yersinia*, it has been suggested to be linked with the phage life cycle[43]. Together with the other additional ORFs (and deletion of less efficient ones), it could provide an improved infection cycle. Indeed, when studied over all bacterial isolates, the host generalist phages isolated most recently were most infective (Supplementary Fig. 3). This likely results from a combination of phage characteristics, but further genetic studies are needed for confirmation. From an evolutionary perspective, parasite host range is expected to correlate negatively with fitness[44]. Our finding of parallel increases in host range and infectivity demonstrates that the evolutionary trajectories of the phage–bacterium interactions are not yet fully understood.

Genome and gene family expansions have been shown to be the starting points of evolutionary innovations in eukaryotic parasites[45]. These gene family expansions are often located in the recombinogenic parts of the genome, which could also apply to the phage genomes in our data set, as the additional ORFs are found upstream of ORF66, which contains a putative ICE domain. ICEs integrate in chromosomes and mediate horizontal gene transfer in bacteria[46], but their possible functionality in viruses has not been characterized. Nevertheless, these phage genome expansions indicate the role of the surrounding environment as a source of novel DNA and highlight the importance of long-term environmental sampling in understanding phage evolution. Yet, as the physical characteristics of the capsid strongly constrain phage genome size, our data raise new questions on what limits the phage host range, virulence, and genome evolution, and what the associated trade-offs are.

The presence of CRISPR-Cas systems has been found in ~40% of studied bacterial species[40], and the molecular mechanisms of different CRISPR systems are under active research[47]. Some repeat-spacer arrays remain relatively constant over millennia[48], while in some species no two identical CRISPR loci are found even in cells living in close proximity[25]. However, previous

CRISPR studies have mainly concentrated on unraveling the mechanisms of CRISPR immunity in laboratory conditions, whereas the role of CRISPR immunity for microbial evolutionary ecology in nature is poorly understood.

Studies on a related bacterium, *Flavobacterium psychrophilum*, have discovered inactive CRISPR systems across multiple strains[49], but also high variability between metagenomic samples from nature[50]. In *F. columnare*, we report two active CRISPR loci: C1, a type II-C system and C2, a recently identified RNA-targeting type VI-B system. Two ORFs downstream of C2, whose translations show possible transposase motifs, may be remnants from possible horizontal transfer of this locus, see e.g., ref. [40].

Both of the CRISPR loci accumulate novel (mostly phage-targeting) spacers extending the backbone of conserved (not phage-targeting) spacers (Fig. 2a). On several occasions, the introduction of CRISPR spacers to the bacterial population was followed by the appearance of phage isolates with modifications in the corresponding protospacer regions. This is consistent with studies demonstrating the appearance of CRISPR-escape mutants due to altered protospacers[5, 17] or PAM regions[51]. In the four most specific protospacer alterations (Fig. 2c), the changes were non-synonymous, leading to alternative amino-acid sequences in the predicted ORFs. While promoting evasion of host CRISPR immunity, these mutations are also likely to have a fitness cost for the phage. Assuming these non-synonymous mutations arose before synonymous ones by chance, a possible explanation for their prevalence may be the large benefit gained from CRISPR-evasion. In this scenario, the costs of these mutations would only be unmasked after the disappearance of CRISPR-based selection pressure, leading to the re-emergence of the ancestral protospacer sequence. However, our data in this respect are limited and these predictions require further investigation.

Spacer acquisition is still one of the least known aspects of CRISPR[52]. We found an unequal distribution of protospacers in phage genomes, indicating that CRISPR defense targets the terminal end of the phage genomes. A recent study showed that in the type II-A CRISPR system of *Streptococcus pyogenes*, spacer acquisition begins while phage dsDNA is being injected inside the cell[53]. This results in polarization of spacer acquisition, with most spacers originating from the phage genome end that first enters the cell. This has been suggested to enrich spacers targeting this region during selection and less efficient interference by spacers targeting the opposite end. Typically, in tailed phages (excluding T4) the end of the packaged genome is the first end to be ejected[54]. While the packaging and injecting direction of the phage genomes in our data set is unknown, protospacer distribution indicates similar polarization as in *S. pyogenes*. Interestingly, this genome end also contains most of the variation among the phages, providing a possible link between the proposed model of spacer acquisition and the observed patterns of phage genome reorganization.

The size of the variable ends of the CRISPR loci varied between the years (Fig. 2a). Surprisingly, the most recent and resistant bacterial isolates (originating from rainbow trout) contained the least number of phage-targeting spacers. While the bacterial isolation source did not significantly affect the resistance patterns (Fig. 2b), it is possible that these bacteria have experienced more phage encounters during bacterial disease outbreaks, which would influence the resistance mechanisms and patterns. It is also possible that the absolute number of spacers is not a decisive factor in determining resistance and that spacers vary in effectiveness based on their target sequence (see e.g., ref. [41]).

Despite the coevolutionary dynamics observed between spacer content and phage genomes, CRISPRs could not completely explain the resistance patterns of the host, possibly due to high phage pressure in the experimental setup[55]. However, under low-

phage selection (such as natural aquatic settings) CRISPRs may still be preferred over mutation-based immunity due to their minimal effects on overall fitness of the host[30], at least when not constantly expressed[56]. Consistent with a previous suggestion[57], the presence of the high diversity of CRISPR spacers (e.g., in 2010) could have reduced phages' ability to overcome bacterial resistance by point mutations, and may have caused larger shifts in the structure of the phage population, observed here as phages with increased genome size. In addition, adsorption percentages in infectious and non-infectious phage–bacterium pairs showed no clear correlation, suggesting that adsorption does not guar-antee infection. Considering the costs related to maintaining broad resistance against phages, combining constitutive immu-nity and diverse CRISPR immunity may be the most successful strategy for a natural bacterial population[58], as the phage have the potential to rapidly overcome these bacterial defense mechanisms to ensure their own persistence. The expanding host range driven by increased genome size is likely to be restricted by the physical capacity of the capsid. Nevertheless, other factors, such as novel anti-CRISPR functions, that limit the evolutionary potential of this specific phage–bacterium interaction have yet to be dis-covered. Furthermore, the expansion of the host range may trade-off with phage infectivity[44], which might be the point at which the adaptive CRISPR defense alone would be sufficient to overcome those infections.

The bacterial and phage isolates used in this study originate from a fish farming environment, a semi-natural system with a constant flow of incoming environmental water. It is possible that although attempts to isolate *F. columnare* phages from waters outside fish farming have been mainly unsuccessful[37], phages and bacteria may have already interacted in nature, and the farming system enriches the phage population sizes to detectable levels. However, as a man-made environment, the phage–bacterium interaction is subjected to selection by farming practices, espe-cially high fish densities and the use of medication, which may select for the most virulent and antibiotic tolerant *F. columnare* strains[59]. Thus, by providing essential information on the phage–bacterium interactions under multifactorial selection in these settings, our data and study system are directly relevant for sustainable use of phage therapy applications in field conditions. For example, in laboratory conditions, eliciting phage resistance in *F. columnare* has been shown to cause reversible morphotypic changes that can maintain constitutive resistance[60, 61]. These resistant morphotypes are, however, missing from natural iso-lates, probably because of their impaired growth, tolerance to protozoan predation and capacity to infect fish[60, 62]. The absence of resistant morphotypes indicates higher benefits of CRISPR immunity under multifactorial selection in field conditions, although our data show that both constitutive and adaptive bacterial resistance mechanisms are likely to be active and important in natural settings.

## Methods

**Phage and bacterial isolates used in this study**. Phage and bacteria were isolated from a private fish farm (from fish and tank water) in Central Finland and from its immediate surroundings from inlet and outlet water (Supplementary Tables 1 and 2). The isolates were originally collected for other purposes, therefore we do not have isolates from each year, and from same source, and also different enrichment hosts were used for phage isolation over the years. The interaction between *F. columnare* and its phages is genotype-specific (each phage type infecting strains of only one bacterial genotype[37]), which allowed us to monitor the evolution of phage–bacterium relationship in this study using a specific host genotype.

Bacterial strains were originally isolated from water samples and from the fish by plating 100 μl of water or gill swabs on Shieh agar[63] or Shieh agar supplemented with tobramycin[64], as previously described[65]. The plates were incubated in room temperature for 48 h, after which individual yellow rhizoid colonies exhibiting the typical colony morphology of *F. columnare* were picked, pure-cultured, and stored frozen at −80 °C, with 10% fetal calf serum and 10% glycerol. Among the collected

bacterial isolates over the years from this farm, bacteria belonging to genetic group C were chosen based on ARISA genotyping using the methodology described previously[65, 66]. In brief, the intergenic spacer region (between the 16S and 23S ribosomal genes) of *F. columnare* isolates was amplified by PCR. The products were analyzed using ABI 3130*xl* Genetic Analyzer to assign the isolates into previously determined genetic groups.

Phages were isolated from water samples by enrichment. A filtered water sample (pore size 0.45 μm, Nalgene) was used to dilute five-fold Shieh medium, and an overnight-grown enrichment host (belonging to genetic group C) was inoculated in this medium. Cultures were grown at room temperature (23 °C), at 110 r.p.m. on a benchtop shaker (New Brunswick Scientific) until they turned turbid. A double agar layer method was used for detecting phages: 300 μl of turbid sample with 3 ml of 0.7% soft Shieh agar (tempered to 47 °C) was applied on solid Shieh agar. After incubation (23 °C) for 24–48 h (depending on the bacterial growth) plaques were picked. Three rounds of plaque purification were performed for each phage isolate. For further characterization, phage lysates were prepared adding 5 ml of culture media on a plate with confluent lysis, and shaken at 8 °C, at 95 r.p.m. on a benchtop shaker for 6 h. Phage isolates were stored frozen in −80 °C with 20% glycerol. Phages were characterized by genome restriction profiles using prior EcoRI experiments, and for this study we used all the phage isolates infecting the bacterial genetic group C.

**Phage host range**. For analyzing phage infectivity and host range, a fresh phage lysate was prepared using host strain C4, which is a non-virulent mutant of *F. columnare*[67]. An overnight-grown bacterial culture (in Shieh medium, 23 °C, 110 r.p.m.) of the test strains was plated using the double agar overlay method by mixing bacterial culture 1:10 with 0.7% soft Shieh agar tempered to 47 °C and pouring this on agar plates. The phage was diluted until million fold ($10^{-6}$), and 10 μl samples of each dilution was spotted on top of the bacterial lawn. All bacterial strains were infected with all phages in a pairwise manner. After incubation of 48 h, the plates were checked for single plaques. Plaque formation was considered as a positive result for phage infectivity.

**Time-shift experiment and statistical analysis**. Bacterial resistance in response to phages from past, contemporary, and future time points was analyzed with generalized linear mixed models using SPSS 22.0. This was done using the infection data presented in Fig. 2, and we calculated the proportion of resistance of bacteria in relation to the temporal difference of isolation of each phage. In the statistical analysis, the response variable of phage infection was encoded as a binary trait, therefore binomial distribution with logit link transformation was used. Time shift was used as a fixed factor and phage isolation year, bacterial isolation place and phage enrichment host as random factors, with bacterial identity as a blocking factor (subject). Only phages able to produce plaques in the host bacteria were considered infective in the analysis.

**Adsorption assay**. Adsorption efficiency of eight phages to nine bacterial hosts (Supplementary Fig. 1) during 20 min was measured following the method by Kropinski 2009[68] with modifications. In brief, bacteria were grown to exponential phase and diluted to ~3×$10^8$ colony forming units ml$^{-1}$. A total volume of 3 ml was used in the assay: 0.3 ml of phage suspension (adjusted to 1–3×$10^5$ plaque forming units (PFU) ml$^{-1}$) was added to 2.7 ml of bacterial culture. Phage added to Shieh medium without bacteria was used as a control. The assays were done in three replicates. After 20 min a sample of 50 μl was diluted to 950 μl of cold growth media supplemented with three drops of chloroform. PFU ml$^{-1}$ of the free phage particles was determined by plating 100 μl of the sample with C4 strain using the double agar overlay method. Adsorption data were analyzed using a generalized linear mixed model with adsorption percentage as the response variable (binomial distribution and logit link) using R 3.3.3 and RStudio 1.0.136. Time shift (past, contemporary, and future phage) and bacterial resistance were used as explanatory variables and replicate identities as random factors. In the few cases where the observed number of plaques in VK20 exceeded the control values due to mea-surement error due to instrumental or observational error (i.e., having 110 plaques resulting from a bacterial adsorption tube, when only 100 plaques were extracted from the control tube), the adsorption percentage was fixed to 0.

**Phage genome sequencing**. Phages subjected to genome sequencing were chosen based on their infection profiles (Fig. 2). Phage DNA was isolated using a protocol described by Santos[69] with slight modifications. Briefly, after RNase (10 μg ml$^{-1}$) and DNase (1 μg ml$^{-1}$) treatment phages were precipitated with 40 mM ZnCl$_2$ and incubated for 5 min followed by pelleting (15,000 × *g*, 5 min) and resuspended in TES buffer (0.1 M Tris-HCl, pH 8; 0.1 M EDTA; 0.3% SDS). DNA was purified using a GeneJET Genomic DNA isolation kit column (Fermentas). About 100 ng of DNA was subjected to genome sequencing with Illumina MiSeq in Institute for Molecular Medicine Finland (FIMM), apart from isolate FCV-1, which was sequenced using Roche 454 at LGC Genomics, Germany.

**Genome assembly**. The phage genomes were assembled using Velvet-assembler (v. 1.2.10)[70] with optimal k-mers per genome (average coverage ~×1600). Differ-ences appearing in only one genome were checked and corrected if needed. Single

insertions and nucleotide differences were checked by sequencing directly from the genome using specific primers (Supplementary Table 7). ORF's were predicted using GeneMarkS (v. 4.28)[71] and Glimmer (v. 3.02b)[72]. Protein Homology/analogY Recognition Engine V 2.0 (PHYRE[2])[73] was used for predicting protein structures.

**CRISPR sequencing and analyses.** Bacterial DNA was extracted from overnight-grown turbid cultures using the GeneJET Genomic DNA isolation kit (Fermentas). We used the previously published *F. columnare* ATCC 49512 genome[39] and our own unpublished sequence data as references in designing primers for amplifying the CRISPR loci. NetPrimer (Premier Biosoft) was used for the analysis. Primers (Supplementary Table 9) were provided by Sigma. CRISPR loci from the extracted DNA were amplified in a PCR reaction (PIKO Thermal cycler by Finnzymes) using the Phusion Flash II polymerase in a 20 µl reaction with a template concentration of 0.5 ng µl$^{-1}$. The protocol was completed following the manufacturer's instructions. The annealing temperature was 65 °C for CRISPR1 and 67 °C for CRISPR2, and the elongation step was 30 s for CRISPR1 and 27 s for CRISPR2. The reactions were purified for sequencing using the QIAquick PCR Purification Kit (Qiagen). The amplified PCR products were sequenced with the Sanger method. Raw data were transformed using SequenceAnalysis 6 basecalling. Two replicates were done for each read and the consensus sequences were manually determined using Geneious 9.1.4 (Biomatters Ltd.).

The determination of the orientations of the CRISPR loci (namely, the transcription direction of the repeat-spacer arrays) was based on analyses using CRISPRDetect[74], comparisons with other similar CRISPR systems and observations of the polarization of spacer acquisition. Unlike in most CRISPR systems, the crRNA in type II-C systems is transcribed starting from the conserved end[75]. By assuming this transcription direction, we could detect a conserved PAM sequence downstream of C1 protospacers. As all type II systems described so far exhibit downstream PAMs, we assume that the C1 repeat-spacer array is also transcribed starting from the conserved end. Very low confidence values associated with C1 direction prediction by CRISPRDetect led us to not use this tool with this locus. However, CRISPRDetect was able to produce confident direction prediction values for the C2 repeat-spacer arrays, which are in line with those of a previous study[31] and which place the transcription initiation point at the variable end, as in most CRISPR systems. This direction also enables the spacers to target messenger RNA (mRNA) from predicted viral ORFs and produces PFSs in line with those of a previous study[31].

All spacers across all isolates were pooled and labeled according to their CRISPR locus, creating global spacer pools for both loci. Spacers were numbered starting from the conserved end of the oldest strain, B425 (in which the oldest C1 spacer is called C1s1 and the most recent one C1s37). After labeling all B425 C1 spacers, we proceeded to the next strain (B447) and searched for any novel spacers, labeling them by adding to the global C1 numerical sequence (e.g., C1s38). Eventually, each C1 and C2 spacer in the total spacer pool had a locus-specific ID. Each spacer may be isolate-specific (e.g., C2s29) or shared by two or more isolates (e.g., C1s48). In C1, a 114 bp area containing degenerate repeats between spacers 4 and 5 was skipped, and numbering was resumed downstream of this area.

The C1 PAMs are reported using the guide-centric approach[76]. Due to the RNA-targeting nature of C2, the PFSs reported here are obtained from the single-stranded mRNA of the predicted ORFs using the target-centric approach.

**Protospacer location analysis.** Possible bias in the distribution of protospacers on the phage genomes was analyzed by comparing the actual protospacer location means against simulated sets of protospacers distributed randomly and uniformly over a 50,666 bp long genome (the size of the consensus sequence). The mean protospacer positions (18 spacers for C1, 15 spacers for C2) of each of these permutations (100,000 simulations) was used as the statistic for creating a null distribution. The observed means from the collection of 18 C1 (mean position 35706.61) and 15 C2 (mean position 33342.33) protospacers were compared to this distribution and *p*-values calculated as the proportion of the simulated means exceeding the observed ones over the total number of simulations. These analyses were conducted with R 3.3.3 and RStudio 1.0.136.

It must be noted that the possible effects of primed spacer adaptation have not been taken into account in these analyses. Priming has not been shown to affect spacer acquisition beyond class I systems[52] and by current knowledge requires the cascade complex and the Cas3 nuclease[77], which are missing from the CRISPR loci of *F. columnare*.

**Analysis.** Geneious (v. 9.1.4) was used for the annotation and analysis of the CRISPR sequence data, and phage PAM and PFS extractions. WebLogo (v. 3.5.0) was used to create sequence logos of the PAM and PFS sequences.

**Data availability.** Phage genome sequences are available in GenBank (accession numbers KY951963, KY951964, KY979235-KY979247, KY992519, KY992520). All other data are included in the supplementary files and available from the authors.

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

## Acknowledgements

This work was supported by the Finnish Centre of Excellence Program of the Academy of Finland; the CoE in Biological Interactions 2012–2017 (#252411), by the Academy of Finland (grants #266879 and #304615), and by the Jane and Aatos Erkko Foundation. We would like to thank MSc Jenni Marjakangas, MSc Katja Neuvonen, and Mr. Petri Papponen for skillful help in the laboratory, Dr. Heidi Kunttu for kindly donating genotyped bacterial strains, Dr. Reetta Penttinen for help in bacterial isolation, and Dr. Andrés López-Sepulcre for help in statistical analyses. In memory of Prof. Jaana Bamford.

## Author contributions

E.L.: Performed the host range experiments and phage genome analyses. V.H.: Sequenced and characterized the bacterial CRISPR loci. J.R.: Assembled the phage genome data and supervised bioinformatics analyses. L.-R.S.: Performed the time shift and statistical analyses and wrote the first draft of the manuscript. All authors participated in data analyses and in writing the manuscript.

## Additional information

**Competing interests:** The authors declare no competing financial interests.

