## [Peer Review File · Nature Communications]

Reviewers' comments:

Reviewer #1 (Remarks to the Author):

Laanto et al. collected bacteria and phages from an industrial fish farm in Finland over the course of seven years, and using time-shift assays they demonstrate that the bacteria and phage coevolve with arms race dynamics. Next, they sequence many of the phage isolates and the CRISPR loci in the bacterial hosts to try to tease apart the molecular mechanism underlying the observed coevolution.

The study presents an interesting dataset in a field with few studies from semi-natural systems and, to my knowledge, none of this duration. However, a major weakness of the study is that the role of CRISPR and surface resistance in the observed coevolutionary dynamics is insufficiently demonstrated. Given the importance of these results, and the paucity of systems with which such questions can be addressed, this manuscript should be accepted pending major revisions.

Major issues:

The first experiment shows the results of a time shift experiment, which demonstrates phage evolution. However, the authors should also test in a similar assay whether hosts evolve (i.e. exposing bacteria from past present and future to contemporary phage).

Second, the authors use phage that has been enriched on different hosts (Table S2). The authors should address how this may influence the results, especially since the host used seems to correlate with the time where phage was isolated. I would like to see a time shift experiment where all phages have been enriched / amplified on the same host genotype.

Another major concern with the dataset is the disparity in sampling sites across the years. These bacterial and phage samples may not have been collected with this study in mind and I think therefore that a better justification of why these isolates were chosen for the experiments is required. As well as adding transparency to the manuscript I think this would justify the unbalanced design of the sampling regime (presumably for logistical reasons). In a similar vein, have the differences in site been controlled for adequately in the mixed effect model presented? The methods section about the statistical model (lines 345-351) could be clearer- my interpretation is that sampling year was included as a random effect but sampling site was not. Could this bias the results in some way? Given that the latter samples were predominantly isolated from trout, whereas earlier samples were from tanks and other sources, it seems possible that these trout samples have greater resistance to phages due to higher bacterial numbers and therefore more frequent encounters with infective phages. I'd like to see some analysis that shows this uneven sampling isn't driving the result.

My other major concern is that much of the CRISPR related data is purely correlational. The authors state that they observe CRISPR mediated resistance and surface modification (lines 82-85) yet I couldn't see where or how they had phenotypically teased these two resistance

mechanisms apart. Whilst I acknowledge that the patterns of spacer acquisition are consistent with co-evolutionary dynamics, understanding the relative importance of these mechanisms remains a major goal in this field. As it is, a lot of the conclusions concerning the molecular basis of coevolution appear unsubstantiated, and at times confusing. Fig. 2 seems to suggest that many clones have lots of spacers with a perfect match against the phage, yet they are scored as being sensitive in many instances. I would like to see some quantitative EOP analysis (instead of binary scoring of resistance/infectivity) that tests the effect of the number of perfect CRISPR spacer matches on resistance phenotype across all bacterial isolates. In this analysis, the authors could use the C1 strain as a reference. The authors should also directly measure phage adsorption for all bacteria-phage interactions in order to substantiate the claims concerning evolution of surface resistance. Again, the C1 strain could serve as a reference strain.

The authors observe that the phage isolated from later time points has gained additional genes, which is potentially interesting. They strongly suggest that these genes are important determinants of increased host range of these phages, but they provide no data to support this. I think the authors should do more; would it for example be possible to express these phage genes from plasmid and observe the effect on infection by phage with narrow host range? If these genes encode proteins that block host immune responses, this activity should become apparent when expressed from plasmid. Similar analyses have been done to demonstrate antiIRM and antiCRISPR activity (see for example the papers from the Bondy-Denomy and Davidson labs).

Minor issues:

A figure / table that shows the number of perfect matching spacers for each bacteria-phage interaction would be very helpful. The figures in their current form are not clear enough to extract this information (Fig. 2 shows only the variable spacers, not the core, and in Fig. S3 it is unclear what clone the spacers come from). Ideally, this figure would be integrated with the EOP data.

I would suggest moving Fig 3 and 4 from main text to supplemental data; overall I would suggest to focus more on the coevolution side of things, and less on mechanistic aspects, such as PAM sequences.

Line 41- the meaning behind this clause was unclear to me- I think the point is that bacteria-phage coevolution shapes microbial communities, but could be a bit clearer.

Line 68 – Different resistance mechanisms are predicted to be important in different ecological conditions – this needs some references or explanation.

Line 85 – RNA-targeting CRISPR- I think this is potentially a bit strong as the study just shows high sequence homology with type VI-B systems. Perhaps add “putatively”.

Line 92- I think this section would benefit from justification of why these samples were collected- during an outbreak of *F. columnare*?

Line 180- This states that matching spacers fluctuated over time- is it possible to link this result to the broader observation that the co-evolutionary dynamic was more of an arms race than fluctuating selection?

Line 206- states that these results are important for phage therapy. I agree, but would like a bit more explanation i.e. predictions about the longevity of a treatment.

Line 163- This section regarding the PAM analysis does little to add to the manuscript. Whilst these patterns may be of interest to those working with similar systems, I think these additional details actually detract from the main result of the paper.

Additionally, I was curious about whether these data could be linked to other ecological data i.e. years in which *F. columnare* infections were high in the fish farms, and whether this led to greater CRISPR/surface modification resistance.

Lines 134- This sentence (and others regarding the phage genome expansion inc. line 231) is a speculative.

Reviewer #2 (Remarks to the Author):

The authors do a long-term co-evolution experiment using *Flavobacterium columnare* and its phages. They show Crispr spacer acquisition by bacterial isolates over time, and changes in phage genome sequences, also over time. The simple take-home message is that recent bacterial isolates are resistant to phages isolated much earlier in time, and vice versa, that recent phage isolates grow on the older bacterial isolates. The molecular characterization takes this longitudinal co-evolution study considerably beyond most others.

Unfortunately, the manuscript is very difficult to digest, and is not in a state suitable for publication. There is much experimental data presented that is not adequately described or discussed. Consequently, important observations are hard to discern from peripheral data, let alone to evaluate thoroughly. Nevertheless, after struggling through the main text and its data, and through the Supplementary material, this reviewer thinks, that after a comprehensive reorganization, a re-write that includes more analytical thought, this study should have a highly significant impact in the field of host-parasite evolution.

The following is not ranked by importance, and is by no means comprehensive.

19. The term "from the future" reads like a sci-fi novel. I suggest re-phrasing, particularly in the Abstract, where the meaning is not obvious. In general, it may be better to refer to phages that were isolated x years before or after a specific bacterial isolate.

20. Although the authors are generally clear in using "correlation", the way it is used in this sentence strongly implies causality, which is not directly shown. Further, the Discussion backs away from the importance of the Crispr loci.

41. "immunity" does not accurately describe resistance to adsorption. Use of resistance here may be more accurate.

90. What do the authors mean by "genotype-specific manner"?

93-4. Were Myoviridae specifically chosen? If so, why?

Fig. 1. This experiment needs better explanation. Which host(s) was used? What does Fig. 1A show that is not present in Fig. 1B? Is 1A necessary?

112. isolates from 2009 to 2014 or 2007- (line123)?

111-120. All 17 sequenced RISA group C phages are very similar to each other and to FCL-2, which the authors previously characterized (ref 48). Was some a priori intentional screening conducted that is not described here, especially since FCL-2 is classified into RISA group G and was isolated from a different fishery (ref. 28). Has this environment selected for a single dominant phage type, even though the population structure of the host bacterium is said to be complex? More information should be provided in the Introduction.

Five phages are said to show differences in four structural proteins. Are these differences the same in all phages, in which case we should be told what they are in the text, or are they different, in which case a short Supplementary Table should be added. What is the experimental evidence that these are, in fact, structural proteins? Rather than annotating them as similar to other uncharacterized phages, only similarity to known phage structural proteins is meaningful.

Table S4 and especially the horrendous Table S3 serve no purpose and both should be deleted. Those interested will do their own analyses.

122-129. Why are we told about a missing HNH and silent changes in a putative helicase? It merely distracts. What is thought to be important in group 2 phages – especially those lacking structural protein changes - that allows them to grow on ancestral hosts?

137. "necessary" makes this an overstatement.

143-5 and Fig. S2. Both are incomprehensible; please provide some explanation.

156-8. Too speculative for Results.

Fig.4. is a waste of graphic space. The same information is provided in Tables S6 and S7, and the text provides more than an adequate description. In fact, this topic: lines 163-177, seems so tangential to the main thrust of the paper it could be abbreviated to 1-2 sentences at most.

179-197. This section should be one of the most interesting aspects of the study but a useful description is completely lacking. The reader is left having to determine how the authors reach their conclusions (which may be correct).

Table S8. If I am reading this correctly, as there is no explanatory legend to help, the Crispr C1 locus expands from 38 to 59 spacers, and C2 from 18 to 29. The text refers to some spacers being lost and those sequences reappearing in phages isolated at later times. The observation is interesting and it should be described more fully. If nothing else, it implies a fitness cost to the phage in changing from the original protospacer sequence. From a simplistic viewpoint, avoidance of a spacer-induced abortive infection could often occur by using synonymous codons. So why do "revertants" arise?

Table S9 also lacks adequate explanation. What do the titers mean? Even though the Crispr systems can target many loci in the phage genome, several isolates seem to be able to grow well, perhaps even ~normally by relative titer, despite being potentially subject to attack. Do the authors think these phages encode some anti-Crispr functions?

301-7. This gratuitous paragraph should be deleted. It is well-accepted that phage mutation

rates are higher than their hosts, and the arms race described elsewhere in the text belies the "once only" claim.

410. Genome sequences should have already been deposited at Genbank, even if not released. The manuscript should not be accepted without Accession numbers.

Reviewer #3 (Remarks to the Author):

Insights to bacteria-phage interactions in nature are especially important in light of the resurgence of interest in phage therapy as an alternative to antibiotics, but are typically restricted to in vitro studies. Following evolutionary change in specific bacteria and phage lineages in nature has thus far proven challenging but is of critical importance in translating the wealth of experimental results to natural systems.

Laanto and colleagues explore the coevolutionary dynamics at both phenotypic and genomic levels of the fish pathogen, *Flavobacterium columnare*, and its associated phages. They demonstrate that phages evolve wider host ranges over time (infecting both contemporary and ancestral bacterial strains) and that bacteria were generally more resistant to phages from the past than phages from the future. They go further to evaluate the molecular changes underpinning these results, and find that the phage increased its genome size and had mutations in the protospacers associated with acquired bacterial resistance via CRISPR was mechanism, and that the bacterial lineage changed both in terms of CRISPR spacers but also in terms of other known resistance mechanisms, such as surface modification. Together, the work offers an exciting insight into the real world dynamics of bacteria and phages, and suggests a complex coevolutionary process involving multiple resistance mechanisms and counter-adaptation.

My only major concern is in regard to the methods of phage isolation, and the potential biases this might introduce. According to the supplemental table, phages from all time points were isolated by enrichment on a few key bacterial hosts, all of which were from 1997-2010. Given that about 1/3 of these phages were isolated from later than 2010, this means that you may have been preferentially selecting for phages that maintained the ability to infect earlier bacteria types (i.e. that had larger host ranges). For instance, it may be that some phages in the sample from 2014 had lost the ability to infect earlier bacterial types (i.e. had a narrower host range) and therefore would have been excluded from your sampling method. Perhaps one way around this would be to see if the expanded host range result holds if you exclude those that were enriched specifically on 'older' strains? The result that broader host range is correlated with larger genome size would still hold, and is indeed interesting! But I worry about the statement that your data support an ever-increasing host range over evolutionary time, where it might be that you are selectively sampling for narrower host breadth early on (with temporally sympatric enrichment) and wider host breadth later. One way or another, this alternative explanation should either be directly addressed or at least discussed as an alternative.

Minor comments:

- In figure 1 it is unclear why both panels are necessary. Is the top panel just a summary of the bottom panel? I think the bottom panel shows the results beautifully.

- In figure 2, the fact that D seems to be above C is a bit strange.

- Figure S1: it seems to me that phages from the final time point will have been tested on more past bacteria (and therefore have more suitable hosts) than those from the beginning which are tested primarily against future bacteria that they are unlikely to infect. I think a more reasonable test of the question whether phage plaque production increased over evolutionary time would be to measure this trait only on contemporary bacteria each time.

Reviewer #1 (Remarks to the Author):

Laanto et al. collected bacteria and phages from an industrial fish farm in Finland over the course of seven years, and using time-shift assays they demonstrate that the bacteria and phage coevolve with arms race dynamics. Next, they sequence many of the phage isolates and the CRISPR loci in the bacterial hosts to try to tease apart the molecular mechanism underlying the observed coevolution.

The study presents an interesting dataset in a field with few studies from semi-natural systems and, to my knowledge, none of this duration. However, a major weakness of the study is that the role of CRISPR and surface resistance in the observed coevolutionary dynamics is insufficiently demonstrated. Given the importance of these results, and the paucity of systems with which such questions can be addressed, this manuscript should be accepted pending major revisions.

RESPONSE: We would like to thank the reviewer for the positive response. We have addressed all the points that were raised, including adsorption tests and data re-analyses, and hope that the reviewer finds the modifications satisfactory.

Major issues:

The first experiment shows the results of a time shift experiment, which demonstrates phage evolution. However, the authors should also test in a similar assay whether hosts evolve (i.e. exposing bacteria from past present and future to contemporary phage).

RESPONSE: In fact, this is the way the analysis was done; all bacteria were exposed to all phages and all the data in figure 2 was used to calculate this. The figure demonstrates the level of bacterial resistance over all phage-bacterium pairs. As the age difference of the phage-bacterium pairs always stays the same (e.g. 2 years) the bacterial resistance evolution can be deduced from the data. E.g. 75% of the bacterial population are resistant to phages 2 years from the past, indicating the level of resistance evolving during this time. We could interpret the data also the other way - as the level of infectivity (see below), but as we concentrate here more on the resistance mechanisms of the hosts, we feel our presentation describes the results more clearly.

Second, the authors use phage that has been enriched on different hosts (Table S2). The authors should address how this may influence the results, especially since the host used seems to correlate with the time where phage was isolated. I would like to see a time shift experiment where all phages have been enriched / amplified on the same host genotype.

RESPONSE: Yes, it is true that we have used different hosts in the initial isolation of the phages. In the text we have explained (lines 275-279): Phage genome expansions were independent of the enrichment host used in the original phage isolation. An earlier isolated enrichment host was always used (except for first five phages in 2009), however, similar phages could be isolated even when using different enrichment hosts

To minimize host effects, all the phage amplifications after isolation have been done using only one bacterial host, C4, which is a non-virulent mutant of F. columnare (See Kunttu et al 2009, Microbial Pathogenesis). We now state this more clearly in the methods (line 437). Furthermore, this same host was used as a general host in the adsorption tests (see below).

Another major concern with the dataset is the disparity in sampling sites across the years. These bacterial and phage samples may not have been collected with this study in mind and I think therefore that a better justification of why these isolates were chosen for the experiments is required. As well as adding transparency to the manuscript I think this would justify the unbalanced design of the sampling regime (presumably for logistical reasons).

RESPONSE: The reviewer is right, this data was not initially collected to perform the current study, but for other purposes. We agree that this needs to be addressed in the text. We have now added the following explanation in materials and methods (lines 404-411): “Phage and bacteria were isolated from a private fish farm (from fish and tank water) in Central Finland and from its immediate surroundings from inlet and outlet water (Supplementary Tables S1 and S2). Originally, the isolates were collected for other purposes, therefore we do not have isolates from each year, or from the same source, and different enrichment hosts were used for phage isolation over the years.” .

In a similar vein, have the differences in site been controlled for adequately in the mixed effect model presented? The methods section about the statistical model (lines 345-351) could be clearer- my interpretation is that sampling year was included as a random effect but sampling site was not. Could this bias the results in some way? Given that the latter samples were predominantly isolated from trout, whereas earlier samples were from tanks and other sources, it seems possible that these trout samples have greater resistance to phages due to higher bacterial numbers and therefore more frequent encounters with infective phages. I'd like to see some analysis that shows this uneven sampling isn't driving the result.

RESPONSE: We also have now tested the impact of the isolation location of the bacterial host, by implementing it as a random factor in the statistical analysis. The overall statistics for time shift remained significant ($F_{(2, 507)}=15.103, p<0.001$) and similarly to the previous calculation, the phage isolation year included as a random factor in this analysis had a significant effect (Wald $Z=3.061, p=0.002$). However, bacterial isolation place (implemented as a random factor in the same model) did not ($Z=1.399, p=0.162$). We now acknowledge this in results (lines 117-121).

My other major concern is that much of the CRISPR related data is purely correlational. The authors state that they observe CRISPR mediated resistance and surface modification (lines 82-85) yet I couldn't see where or how they had phenotypically teased these two resistance mechanisms apart. Whilst I acknowledge that the patterns of spacer acquisition are consistent with co-evolutionary dynamics, understanding the relative importance of these mechanisms remains a major goal in this field. As it is, a lot of the conclusions concerning the molecular basis of coevolution appear unsubstantiated, and at times confusing. Fig.2 seems to suggest that many clones have lots of spacers with a perfect match against the phage, yet they are scored as being sensitive in many instances. I would like to see some quantitative EOP analysis (instead of binary scoring of resistance/infectivity) that tests the effect of the number of perfect CRISPR spacer matches on resistance phenotype across all bacterial isolates. In this analysis, the authors could use the C1 strain as a reference.

RESPONSE: We agree that the data has a correlational nature. We have now modified the text in respect to the constitutive resistance mechanisms. Surface modifications are only one part of phage

resistance mechanisms and phage infections may be prevented also in the later steps of the infection. We now concentrate in discussing “constitutive defence” instead of “surface modifications”.

We have included the original plaque numbers (removed from supplementary table 9 in the initial submission) and an EOP analysis (with the B230 bacterial host) in the supplementary file (Supplementary Table S3). Unfortunately, we are no longer able to use the C1 strain for the analyses. This strain was originally isolated by the Finnish Food Safety Authority EVIRA, and we had the permission to use the strain until 2013, after which we had to remove the strain from our collections.

While EOP (using strain B230 as a reference host) and number of identical spacers was significantly correlated (Spearman’s rho -0.234, $p=0.002$, $n=169$), there are several reasons we do not think the result is reliable:

1) We cannot rule out the role of constitutive defence mechanisms for this result. For example, V165 and V181 have identical genomes and the equal amounts of spacer hits – yet their infection profiles differ.

2) The spacer content (number and the sequence they target) differs between bacterial strains, and some spacers might be more important than others. It has been shown, for example that the most recent spacer is the most important in *Streptococcus pyogenes* (McGinn & Barraffini 2016, *Molecular Cell*).

3) It would be best to use an experimental system to confirm the role of number of exact spacer matches on phage infectivity. In this experiment we would expose a bacterial strain to phage and let the CRISPR defence evolve. Using evolved bacterial strains with different numbers of spacers we could then test the infection success of phage without influence of variance in constitutive immunity. We are currently doing a separate study that addresses these issues.

With these justifications, we feel that presenting EOP/spacer number data in the manuscript is premature and does not give a true picture of the role on CRISPR immunity on phage infectivity.

The authors should also directly measure phage adsorption for all bacteria-phage interactions in order to substantiate the claims concerning evolution of surface resistance. Again, the C1 strain could serve as a reference strain.

RESPONSE: This is a very important comment for exploring the constitutive resistance mechanisms, whether it is due to surface modification or some other mechanisms targeted towards a later step in phage infection cycle. To tease these mechanisms apart, we have now performed the adsorption experiments the referee suggested. However, we feel it is unnecessary to perform adsorption tests for all replicates, because majority of phage-bacterium pairs produce infection, therefore we know they adsorb. We focused in studying the adsorption of eight of the sequenced phages isolated from all time points in later isolated bacterial hosts to detect surface resistance.

The results are now presented in the text (lines 123-139) and in Supplementary Figure S1. We found that the phage adsorption varies greatly for each bacterium and for most phage-bacterium pairs it is very speculative to say whether the lower adsorption efficiency is the result of surface resistance. However, we found that the phages adsorbed significantly less to bacteria isolated in later time points compared to the earlier isolates, which points to the direction that some surface modifications could exist. Yet, we feel that each phage-bacterium pair needs to be examined also individually to learn the details of the resistance mechanisms.

But as said, we have now removed the claims of surface resistance evolution and prefer using the term “constitutive defence”. We hope that with these changes are sufficient to alleviate our claims.

The authors observe that the phage isolated from later time points has gained additional genes, which is potentially interesting. They strongly suggest that these genes are important determinants of increased host range of these phages, but they provide no data to support this. I think the authors should do more; would it for example be possible to express these phage genes from plasmid and observe the effect on infection by phage with narrow host range? If these genes encode proteins that block host immune responses, this activity should become apparent when expressed from plasmid. Similar analyses have been done to demonstrate antiRM and antiCRISPR activity (see for example the papers from the Bondy-Denomy and Davidson labs).

RESPONSE: This is an excellent idea and would test the effect of the extra DNA in the phage genomes efficiently. Unfortunately, we are unable to perform such experiments, because a functional plasmid system for F. columnare has not been established. We have had multiple attempts to maintain plasmids in F. columnare, but these experiments have failed, as the cells die even in presence of antibiotic selection. There are other groups also working with this issue, with similar difficulties.

We have now acknowledged the need for further genetic studies to confirm the increased infectivity and host range in discussion (line 288).

Minor issues:

A figure / table that shows the number of perfect matching spacers for each bacteria-phage interaction would be very helpful. The figures in their current form are not clear enough to extract this information (Fig. 2 shows only the variable spacers, not the core, and in Fig. S3 it is unclear what clone the spacers come from). Ideally, this figure would be integrated with the EOP data.

RESPONSE: While the number of perfectly matching spacers in each bacterium-phage interaction is already shown in Figure 2B, we agree that the overall presentation of spacers and their hits is ambiguous. Only the variable spacers are shown in Figure 2, because the core set of sequences shared by all bacterial isolates do not contain any known phage-matching spacers. To clarify the CRISPR targeting in the phage genomes further, we have now made Supplementary Figure S5 to show the ID of each phage-targeting spacer. Here, we also highlight the areas where phage genomes differ in their protospacer areas. This figure combined with Figure 2A/B will allow more convenient analysis of phage-bacterium relationships down to the level of individual spacers. We have also improved the explanation of the spacer numbering scheme in Materials and Methods.

We have included the plaque data in the supplementary file, as original plaque counts (Supplementary Table S3A), and as EOP counts with strain B230 as reference strain (Supplementary Table S3B).

I would suggest moving Fig 3 and 4 from main text to supplemental data; overall I would suggest to focus more on the coevolution side of things, and less on mechanistic aspects, such as PAM sequences.

RESPONSE: We have now done a major revision on the text, and focus more on coevolution. Although the main results of e.g. PAMs and spacer targets are still retained in the main text, the details of these findings have been moved to Supplemental Discussion. We have moved Figure 3 to Supplementary Figure S4 and removed Figure 4 and combined this data to Supplementary Tables S9 and S10

Line 41- the meaning behind this clause was unclear to me- I think the point is that bacteria-phage coevolution shapes microbial communities, but could be a bit clearer.

RESPONSE: We now state (lines 41-43): "This has been shown especially under experimental settings, where lethal infections by bacterial viruses, (bacterio)phages, shape the diversity and dynamics of the coevolving host bacterial populations"

Line 68 – Different resistance mechanisms are predicted to be important in different ecological conditions – this needs some references or explanation.

RESPONSE: The following references were added: Westra et al. 2015, Hamilton et al. 2008, Hout et al. 2016.

Line 85 – RNA-targeting CRISPR- I think this is potentially a bit strong as the study just shows high sequence homology with type VI-B systems. Perhaps add “putatively”.

RESPONSE: We now state (lines 96-97): This is the first study to demonstrate a type VI-B CRISPR system functioning in its natural host and in a natural setting.

Line 92- I think this section would benefit from justification of why these samples were collected- during an outbreak of *F. columnare*?

*RESPONSE: We have put effort in clarifying the sampling (lines 101-108). In fact, the sampling does not concentrate only to outbreak periods, but samples have been collected also during other times. However, *F. columnare* has only been isolated from the farm during the warmest time of the year. Interestingly, there is one phage (VK58) in the dataset, which has been isolated during October.*

Line 180- This states that matching spacers fluctuated over time- is it possible to link this result to the broader observation that the co-evolutionary dynamic was more of an arms race than fluctuating selection?

RESPONSE: We have rephrased this sentence and omitted the word “fluctuated” (lines 215-217), as spacers did not in fact show obvious fluctuation (although, in some cases, identical spacers did reappear among separate host strains over time).

Line 206- states that these results are important for phage therapy. I agree, but would like a bit more explanation i.e. predictions about the longevity of a treatment.

RESPONSE: We have now added this point in the introduction (lines 77-78): “From applied perspective, such information is also crucial for the development phage therapy applications²⁶, where evolution of resistance is expected to limit the functionality of the treatment over time.”

Line 163- This section regarding the PAM analysis does little to add to the manuscript. Whilst these patterns may be of interest to those working with similar systems, I think these additional details actually detract from the main result of the paper.

RESPONSE: This section has now been shortened considerably (lines 207-210) and is discussed further in Supplemental Discussion.

Additionally, I was curious about whether these data could be linked to other ecological data i.e. years in which *F. columnare* infections were high in the fish farms, and whether this lead to greater CRISPR/surface modification resistance.

RESPONSE: Interesting idea! Using this dataset this question is, unfortunately, impossible to answer. There are several genetically different bacterial strains co-occurring at the farm, and we have not yet gathered enough phage isolates infecting all bacterial types. Furthermore, disease outbreak frequency is connected with high water temperatures (see e.g. Karvonen et al., 2010 Int. J. Parasitol.) which varies between the years, and undoubtedly the use of antibiotics at the farm also influence the phage-bacterium dynamics in some ways. Perhaps in few years time we would have enough data to look at the data from a larger perspective.

Lines 134- This sentence (and others regarding the phage genome expansion inc. line 231) is a speculative.

RESPONSE: The sentence on line 134 has been removed. We also slightly modified the addressed sentence on line 231 (now lines 298-300), while acknowledging the need for further genetic studies (line 287).

Reviewer #2 (Remarks to the Author):

The authors do a long-term co-evolution experiment using *Flavobacterium columnare* and its phages. They show Crispr spacer acquisition by bacterial isolates over time, and changes in phage genome sequences, also over time. The simple take-home message is that recent bacterial isolates are resistant to phages isolated much earlier in time, and vice versa, that recent phage isolates grow on the older bacterial isolates. The molecular characterization takes this longitudinal co-evolution study considerably beyond most others.

Unfortunately, the manuscript is very difficult to digest, and is not in a state suitable for publication. There is much experimental data presented that is not adequately described or discussed. Consequently, important observations are hard to discern from peripheral data, let alone to evaluate thoroughly. Nevertheless, after struggling through the main text and its data, and through the Supplementary material, this reviewer thinks, that after a comprehensive reorganization, a re-write that includes more analytical thought, this study should have a highly significant impact in the field of host-parasite evolution.

RESPONSE: We thank the reviewer for positive response and the constructive criticism. As suggested, we have cleaned the supplementary data and carefully gone through the text, and re-written parts of results and discussion. We now concentrate more on the evolutionary aspects of the data.

The following is not ranked by importance, and is by no means comprehensive.

19. The term “from the future” reads like a sci-fi novel. I suggest re-phrasing, particularly in the Abstract, where the meaning is not obvious. In general, it may be better to refer to phages that were isolated x years before or after a specific bacterial isolate.

RESPONSE: We have edited the abstract according to the reviewer’s suggestion.

20. Although the authors are generally clear in using “correlation”, the way it is used in this sentence strongly implies causality, which is not directly shown. Further, the Discussion backs away from the importance of the Crispr loci.

RESPONSE: We have now removed “correlation” from abstract, and state “This was also associated with expansion in phage genome size”. Discussion on the co-evolutionary dynamics has been significantly modified (lines 320-357).

41. “immunity” does not accurately describe resistance to adsorption. Use of resistance here may be more accurate.

90. What do the authors mean by “genotype-specific manner”?

*RESPONSE: We have performed cross-infection studies with our collection of *F. columnare* strains and phages. Phages tested infect specifically strains of one bacterial genotype (genotype C). We have changed the text to “*F. columnare* phages are genotype-specific, each infecting strains of only one genotype” (lines 408-409).*

93-4. Were Myoviridae specifically chosen? If so, why?

RESPONSE: No, Myoviridae were not specifically chosen. After several years of isolating phages infecting F. columnare, we have received only one member of Podoviridae and all the other isolates characterized thus far have been members of family Myoviridae (See also Laanto et al 2011).

Fig. 1. This experiment needs better explanation. Which host(s) was used? What does Fig. 1A show that is not present in Fig. 1B? Is 1A necessary?

RESPONSE: The original infection data used in this analysis is presented in Figure 2. In short, all bacteria were infected with all phages. Then, samples of phage populations from different time points were tested in combination with samples of bacterial populations from other particular moments in time. We are now more specific how this was done in the methods section (lines 86-87). In addition, we have removed the Figure 1A, as the previous Figure 1B presents the same data in more detail.

112. isolates from 2009 to 2014 or 2007- (line123)?

RESPONSE: All the phage isolates were from years 2009-2014 and bacterial isolates from years 2007-2013. This has now been corrected.

111-120. All 17 sequenced RISA group C phages are very similar to each other and to FCL-2, which the authors previously characterized (ref 48). Was some a priori intentional screening conducted that is not described here, especially since FCL-2 is classified into RISA group G and was isolated from a different fishery (ref. 28). Has this environment selected for a single dominant phage type, even though the population structure of the host bacterium is said to be complex? More information should be provided in the Introduction.

RESPONSE: There was no a priori genomic screening involved. In fact, some of the phage genomes used in the current study were sequenced at the same time as FCL-2. Furthermore, as the FCL-2 phage infects RISA group G and not C, we did not expect to find such close similarities with these genomes. We have several bacterial isolates and genotypes available from different farms. Although the farming conditions can select for certain genotypes, we do not have enough temporal data for phages that we could say a certain type dominates. Yet, from this particular farm, we have isolated also phages that differ in capsid size, tail length and genome size from the ones we present here. The population structure of F. columnare is in fact not very complex (Ashrafi et al. 2015). Our use of the word "complex" on line 47 referred to the overall biotic and abiotic interactions present in fish farms. In this paper we focus on only a single genotype and its phages and therefore do not consider necessary to speculate on other genotypes and their phages.

Five phages are said to show differences in four structural proteins. Are these differences the same in all phages, in which case we should be told what they are in the text, or are they different, in which case a short Supplementary Table should be added. What is the experimental evidence that these are, in fact, structural proteins? Rather than annotating them as similar to other uncharacterized phages, only similarity to known phage structural proteins is meaningful.

RESPONSE: The reviewer is correct; this was not clearly explained in the text. We have now added a more specific description of the changes in the Supplementary Discussion. The changes in ORFs 27 (one amino acid difference E -> K) and 28 (two amino acid differences, both V -> I) are identical in two phages (V156 and V157) and the changes in ORFs 36 (several differences, including four additional amino acids and one missing) and 37 (three amino acid changes A -> V, K -> R and N -> H) are identical between the three phages (V175, V181 and V182). Unfortunately, the annotation table (Table S2 that has now been removed) included only the best hit (E-value) and did not show other hits that were to known phage proteins. ORF 27 was annotated as a structural protein because of several hits to tail proteins and also a hit to Mu-like prophage tail sheath protein gpL domain (see Table S1). ORF 28 was annotated as a structural protein based on a hit to Cellulophaga phage phiSM

experimentally confirmed structural protein (Holmfeldt et al., 2013)(E-value 6e-45). Because phiSM seems to be closely related, especially in several structural proteins, it was concluded that ORF 27 is a putative structural protein. These results are presented in the Supplementary Discussion and shortly mentioned on lines 151-153.

The reviewer raises a valid concern, as ORFs 36 and 37 do not have any supportive structural protein hits in the database. However, they follow ORFs that contain predicted putative base plate domain (E-value 6.39e-21), tail protein domain (E-value 3.04e-10) and one hypothetical protein (see Supplementary Table S4). As tail genes are in many phages clustered and the genes encoding base plate are in many cases next to several other tail genes, we want to speculate that it is likely that these two putative ORFs are structural (based on the synteny between similar phages). We have now written the annotation results to have a more hypothetical tone. As we have no experimental evidence of any of the predicted ORFs being structural proteins let alone real proteins at all, all annotations have been submitted as “Hypothetical protein”.

Table S4 and especially the horrendous Table S3 serve no purpose and both should be deleted. Those interested will do their own analyses.

RESPONSE: Table S3 has been removed. We agree this had redundant information. However, we see that table S4 is important for interpretation of the data, therefore we have maintained it in the revised version.

122-129. Why are we told about a missing HNH and silent changes in a putative helicase? It merely distracts. What is thought to be important in group 2 phages – especially those lacking structural protein changes - that allows them to grow on ancestral hosts?

RESPONSE: We have now taken out the mentions about HNH and changes in putative helicase. There are also other changes in the genomes of group 2 phages that are identical to the group 3 phages and we suggest that these genomic areas are important for the phage life cycle in ancestral hosts, in other words the phage is able to infect (no changes in structural proteins – no surface modifications inhibiting the first steps of infection cycle) and is also able to use for example the host replication machinery more efficiently because of these changes.

137. “necessary” makes this an overstatement.

RESPONSE: We have now removed this sentence.

143-5 and Fig. S2. Both are incomprehensible; please provide some explanation.

RESPONSE: We have modified the text (now on lines 174-178) and the legend of Figure S3 (previously Figure S2) to make them clearer for the readers

156-8. Too speculative for Results.

RESPONSE: This was transferred into discussion.

Fig.4. is a waste of graphic space. The same information is provided in Tables S6 and S7, and the text provides more than an adequate description. In fact, this topic: lines 163-177, seems so tangential to the main thrust of the paper it could be abbreviated to 1-2 sentences at most.

RESPONSE: We thank for the constructive criticism. Figure 4 has been removed from the main text. In addition, we have significantly reduced text, as the reviewer suggested.

179-197. This section should be one of the most interesting aspects of the study but a useful description is completely lacking. The reader is left having to determine how the authors reach their conclusions (which may be correct).

RESPONSE: We agree that the description of the coevolutionary dynamics needed to be more thorough. We have now completely rewritten this part (lines 213-246, 320-357) to explain our observations in more detail, and significantly expanded the discussion to reflect the significance of these results.

Table S8. If I am reading this correctly, as there is no explanatory legend to help, the Crispr C1 locus expands from 38 to 59 spacers, and C2 from 18 to 29. The text refers to some spacers being lost and those sequences reappearing in phages isolated at later times. The observation is interesting and it should be described more fully. If nothing else, it implies a fitness cost to the phage in changing from the original protospacer sequence. From a simplistic viewpoint, avoidance of a spacer-induced abortive infection could often occur by using synonymous codons. So why do “revertants” arise?

RESPONSE: We have now described these coevolutionary dynamics in more detail on lines 325-334: “In the four most specific protospacer alterations (Figure 2C), the changes were non-synonymous, leading to alternative amino acid sequences in the predicted ORFs. While promoting evasion of host CRISPR-immunity, these mutations are also likely to have a fitness cost to the phage. Assuming these non-synonymous mutations arose before synonymous ones by chance, a possible explanation for their prevalence may be the large benefit gained from CRISPR-evasion. In this scenario, the costs of these mutations would only be unmasked after the disappearance of CRISPR-based selection pressure, leading to the re-emergence of the ancestral protospacer sequence. However, the data in this respect is limited and these predictions require further investigation.”

Table S9 also lacks adequate explanation. What do the titers mean? Even though the Crispr systems can target many loci in the phage genome, several isolates seem to be able to grow well, perhaps even ~normally by relative titer, despite being potentially subject to attack. Do the authors think these phages encode some anti-Crispr functions?

RESPONSE: Table S9 has been removed (phage titers are now presented in Table S3 in a more clear way). We have now covered the possibility of anti-CRISPR proteins in the phage genomes (lines 248-251). However, we could not detect any homologs based on previously discovered type II anti-CRISPR proteins. As these proteins are known to be highly diverse and lacking of easily detectable conserved motifs, it is still possible that F. columnare phages encode their own unidentified anti-CRISPR proteins.

301-7. This gratuitous paragraph should be deleted. It is well-accepted that phage mutation rates are higher than their hosts, and the arms race described elsewhere in the text belies the “once only” claim.

RESPONSE: We have now removed the last paragraph of the text.

410. Genome sequences should have already been deposited at Genbank, even if not released. The manuscript should not be accepted without Accession numbers.

RESPONSE: Phage genomes have been submitted to GenBank and the accession numbers are given in Supplementary Table 2 and at the end of the manuscript.

Reviewer #3 (Remarks to the Author):

Insights to bacteria-phage interactions in nature are especially important in light of the resurgence of interest in phage therapy as an alternative to antibiotics, but are typically restricted to in vitro studies.

Following evolutionary change in specific bacteria and phage lineages in nature has thus far proven challenging but is of critical importance in translating the wealth of experimental results to natural systems.

Laanto and colleagues explore the coevolutionary dynamics at both phenotypic and genomic levels of the fish pathogen, *Flavobacterium columnare*, and its associated phages. They demonstrate that phages evolve wider host ranges over time (infecting both contemporary and ancestral bacterial strains) and that bacteria were generally more resistant to phages from the past than phages from the future. They go further to evaluate the molecular changes underpinning these results, and find that the phage increased its genome size and had mutations in the protospacers associated with acquired bacterial resistance via CRISPR as mechanism, and that the bacterial lineage changed both in terms of CRISPR spacers but also in terms of other known resistance mechanisms, such as surface modification. Together, the work offers an exciting insight into the real world dynamics of bacteria and phages, and suggests a complex coevolutionary process involving multiple resistance mechanisms and counter-adaptation.

RESPONSE: Thank you for the positive response.

My only major concern is in regard to the methods of phage isolation, and the potential biases this might introduce. According to the supplemental table, phages from all time points were isolated by enrichment on a few key bacterial hosts, all of which were from 1997-2010. Given that about 1/3 of these phages were isolated from later than 2010, this means that you may have been preferentially selecting for phages that maintained the ability to infect earlier bacteria types (i.e. that had larger host ranges). For instance, it may be that some phages in the sample from 2014 had lost the ability to infect earlier bacterial types (i.e. had a narrower host range) and therefore would have been excluded from your sampling method. Perhaps one way around this would be to see if the expanded host range result holds if you exclude those that were enriched specifically on 'older' strains? The result that broader host range is correlated with larger genome size would still hold, and is indeed interesting! But I worry about the statement that your data support an ever-increasing host range over evolutionary time, where it might be that you are selectively sampling for narrower host breadth early on (with temporally sympatric enrichment) and wider host breadth later. One way or another, this alternative explanation should either be directly addressed or at least discussed as an alternative.

RESPONSE: C1 has been isolated from another location already in 1997 and it does not seem to select for a narrow host range. B366 used in isolation of 2014 phages has been isolated in 2010. Only cases here are the first five 2009 phages that have been enriched with bacteria sympatric in time (and VK42 in 2010). However, there is no significant difference in phage host range depending on the enrichment host used. Phages FCV10 (B247 isolation host) and FCV11 (C1 host), and phages VK157 (B270) and VK158 (C1 host) have nearly identical host ranges and genomes. We now state this in lines 274-278: "Phage genome expansions were independent on the enrichment host used in the original phage isolation, an earlier isolated enrichment host was always used (except for first five phages in 2009, Table 2 and Supplementary Table 2), and similar phages were isolated using different hosts (e.g. phage pairs VK156 and VK157, and FCV-10 and FCV-11)."

Minor comments:

- In figure 1 it is unclear why both panels are necessary. Is the top panel just a summary of the bottom panel? I think the bottom panel shows the results beautifully.

RESPONSE: Figure 1A has been removed

-In figure 2, the fact that D seems to be above C is a bit strange.

RESPONSE: We have modified the order of letters

- Figure S1: it seems to me that phages from the final time point will have been tested on more past bacteria (and therefore have more suitable hosts) than those from the beginning which are tested primarily against future bacteria that they are unlikely to infect. I think a more reasonable test of the question whether phage plaque production increased over evolutionary time would be to measure this trait only on contemporary bacteria each time.

RESPONSE: We agree. However, we do not have phage and bacterial isolates from same years to perform the test in this way. Therefore, we have not changed the analysis, but state the following in results (lines 177-178): However, it should be noted that here we were not able to test the plaque production against contemporary bacterial isolates, which may influence the results.

Reviewers' comments:

Reviewer #1 (Remarks to the Author):

The authors have addressed all my concerns in a satisfactory manner. They have thoroughly revised the manuscript, which is now more streamlined and focussed. I have no further criticisms, and feel this is a very interesting study that will be well received by the field.

Reviewer #3 (Remarks to the Author):

I think the presentation of text and results is improved, and I remain excited by and supportive of the data. However, I still have a major hesitation about one of the key conclusions being reached by the authors.

To restate my original concern regarding the assertion that phage host range increases over time: the bacterial host used for enrichment of environmental phages is extremely important to all downstream interpretations. Based on coevolutionary theory, especially under fluctuating selection dynamics, we would predict that phages from further and further into the future begin losing their infectivity to older and older bacterial strains (no selection to maintain these infectivity alleles/mutations). Thus, by only 'choosing' phages that are still capable of infecting bacteria from the past, the authors are necessarily tipping the fishing expedition in favor of broad host range phages (i.e. those still capable of infecting bacteria that they may not have 'seen' - in a coevolutionary sense - for many many years). As such, the fact that all phages from 2014 were chosen because of their ability to infect bacterial strains from 2009/2010, whereas those phages isolated in 2009 were isolated on contemporary bacteria (1/2) or those from 1997 (1/2), and 1 phage from 2010 was isolated on a contemporary host, raises the concern that your host range increase is a result of sampling bias.

Given that you have phages from each time point that were isolated on bacteria from much earlier, I would suggest an easy way to rule this out would be to run the analysis only on those phages that were enriched on non-contemporary bacteria. This should still leave you with a good sample size, and it would rule out one of the most obvious alternative explanations for your results. Alternatively, or ideally in addition, you could include host enrichment strain in your model to make sure the results aren't driven by bacteria C1 selecting for phages with more narrow host range and bacteria B366 selecting for phages with more broad host ranges (and/or perhaps even larger genomes?)

The authors state "C1 has been isolated from another location already in 1997 and it does not seem to select for a narrow host range." If the above analysis cannot be done, I would at least suggest showing this result, although I would much prefer to see the above analyses done, given it should be quite straight-forward. Indeed, the authors state in their rebuttal that "there is no significant difference in phage host range depending on the enrichment host used" but only present anecdotal evidence. This suggests the full analysis should be both very informative (and confirm their conclusions) and doable.

Minor comments:

Line 118: Bacterial isolation place? This is a surprising variable given the earlier text that all bacteria came from the same farm. Even though the different sources are mentioned in the methods, I would briefly mention them here as well.

Supplemental figure 1: This figure seems to show change in absorption rate of phages when tested in a bacterial time shift, but there is no clear indication of what year each bacteria was isolated, nor when each phage was isolated (i.e. where the contemporary time point is) making this very difficult to interpret.

Supplemental figure 3: It is unclear from the methods how these inocula were standardized (e.g. what is the number of plaque forming units produced relative to? Number of known particles in the inocula?) This is important to mention, as otherwise your results only show differences in phage titer - which is not biologically interesting.

Reviewer #3 (Remarks to the Author):

I think the presentation of text and results is improved, and I remain excited by and supportive of the data. However, I still have a major hesitation about one of the key conclusions being reached by the authors.

To restate my original concern regarding the assertion that phage host range increases over time: the bacterial host used for enrichment of environmental phages is extremely important to all downstream interpretations. Based on coevolutionary theory, especially under fluctuating selection dynamics, we would predict that phages from further and further into the future begin losing their infectivity to older and older bacterial strains (no selection to maintain these infectivity alleles/mutations). Thus, by only 'choosing' phages that are still capable of infecting bacteria from the past, the authors are necessarily tipping the fishing expedition in favor of broad host range phages (i.e. those still capable of infecting bacteria that they may not have 'seen' - in a coevolutionary sense - for many many years). As such, the fact that all phages from 2014 were chosen because of their ability to infect bacterial strains from 2009/2010, whereas those phages isolated in 2009 were isolated on contemporary bacteria (1/2) or those from 1997 (1/2), and 1 phage from 2010 was isolated on a contemporary host, raises the concern that your host range increase is a result of sampling bias.

Given that you have phages from each time point that were isolated on bacteria from much earlier, I would suggest an easy way to rule this out would be to run the analysis only on those phages that were enriched on non-contemporary bacteria. This should still leave you with a good sample size, and it would rule out one of the most obvious alternative explanations for your results. Alternatively, or ideally in addition, you could include host enrichment strain in your model to make sure the results aren't driven by bacteria C1 selecting for phages with more narrow host range and bacteria B366 selecting for phages with more broad host ranges (and/or perhaps even larger genomes?)

The authors state "C1 has been isolated from another location already in 1997 and it does not seem to select for a narrow host range." If the above analysis cannot be done, I would at least suggest showing this result, although I would much prefer to see the above analyses done, given it should be quite straight-forward. Indeed, the authors state in their rebuttal that "there is no significant difference in phage host range depending on the enrichment host used" but only present anecdotal evidence. This suggests the full analysis should be both very informative (and confirm their conclusions) and doable.

RESPONSE: We understand the concern of the reviewer. We have now included the phage enrichment host as a random factor in the GLMM model, as the reviewer suggested (lines 118-120). Enrichment host did not significantly affect the infection ($Z=0.586$, $p=0.558$). To confirm this result, we also tested the effect of the enrichment host on phage host range (number of host it infects), and also here the effect of enrichment host is not statistically significant ($\chi^2=11.7$, $df=7$, $p=0.103$). However, this analysis is not included in the main text.

Minor comments:

Line 118: Bacterial isolation place? This is a surprising variable given the earlier text that all bacteria came from the same farm. Even though the different sources are mentioned in the methods, I would briefly mention them here as well.

RESPONSE: An explanation similar to one found in materials and methods has now been added in the results.

Supplemental figure 1: This figure seems to show change in absorption rate of phages when tested in a bacterial time shift, but there is no clear indication of what year each bacteria was isolated, nor when each phage was isolated (i.e. where the contemporary time point is) making this very difficult to interpret.

RESPONSE: We have now indicated the isolation years for both phages and the bacteria in the figures.

Supplemental figure 3: It is unclear from the methods how these inocula were standardized (e.g. what is the number of plaque forming units produced relative to? Number of known particles in the inocula?) This is important to mention, as otherwise your results only show differences in phage titer - which is not biologically interesting.

RESPONSE: We admit that this was unclear. We now present this data differently, using the titers that were standardized to the infections in host B230 (Supplementary Figure 3). This EOP data, together with the original titers, are given in supplementary Table 3. We also reanalyzed the infectivity of phages in different genomic groups, and the results remain significant.

REVIEWERS' COMMENTS:

Reviewer #3 (Remarks to the Author):

This version adequately addresses my concerns.